# Surimi Production from Tropical Mackerel: A Simple Washing Strategy for Better Utilization of Dark-Fleshed Fish Resources

Worawan Panpipat [1], Porntip Thongkam [1], Suppanyoo Boonmalee [1], Hasene Keskin Çavdar [2] and Manat Chaijan [2,*]

1 Food Technology and Innovation Research Center of Excellence, School of Agricultural Technology and Food Industry, Walailak University, Thasala 80160, Thailand; pworawan@wu.ac.th (W.P.); pohntip23@gmail.com (P.T.); suppanyoo2544@gmail.com (S.B.)
2 Department of Food Engineering, Faculty of Engineering, Gaziantep University, TR-27310 Gaziantep, Turkey; hasenekeskin@gantep.edu.tr
* Correspondence: cmanat@wu.ac.th; Tel.: +66-7567-2316; Fax: +66-7567-2302

**Abstract:** Mackerel (*Auxis thazard*), a tropical dark-fleshed fish, is currently a viable resource for the manufacture of surimi, but the optimal washing procedure for more efficient use of this particular species is required right away. Washing is the most critical stage in surimi production to ensure optimal gelation with odorless and colorless surimi. The goal of this study was to set a simple washing medium to the test for making mackerel surimi. Washing was performed three times with different media. T1 was washed with three cycles of cold carbonated water (CW). T2, T3, and T4 were washed once with cold CW containing 0.3%, 0.6%, or 0.9% NaCl, followed by two cycles of cold water. T5, T6, and T7 were produced for three cycles with CW containing 0.3%, 0.6%, or 0.9% NaCl. For comparison, unwashed mince (U) and conventional surimi washed three times in cold tap water (C) were employed. The maximum yield (62.27%) was obtained by washing with T1. When varying quantities of NaCl were mixed into the first washing medium (T2–T4), the yield decreased with increasing NaCl content (27.24–54.77%). When washing with NaCl for three cycles (T5–T7), the yield was greatly decreased (16.69–35.23%). Conventional surimi washing (C) produced a yield of roughly 40%, which was comparable to T3. Based on the results, treatments that produced lower yields than C were eliminated in order to maximize the use of fish resources and for commercial reasons. The maximum NaCl content in CW can be set at 0.6% only during the first washing cycle (T3). Because of the onset of optimal unfolding as reported by specific biochemical characteristics such as $Ca^{2+}$-ATPase activity (0.2 μmol inorganic phosphate/mg protein/min), reactive sulfhydryl group (3.61 mol/$10^8$ g protein), and hydrophobicity (64.02 μg of bromophenol blue bound), T3 washing resulted in surimi with the greatest gel strength (965 g.mm) and water holding capacity (~65%), with fine network structure visualized by scanning electron microscope. It also efficiently removed lipid (~80% reduction), myoglobin (~65% reduction), non-heme iron (~94% reduction), and trichloroacetic acid-soluble peptide (~52% reduction) contents, which improves whiteness (~45% improvement), reduces lipid oxidation (TBARS value < 0.5 mg malondialdehyde equivalent/kg), and decreases the intensity of the gel's fishy odor (~30% reduction). As a result, washing mackerel surimi (*A. thazard*) with CW containing 0.6% (*w/v*) NaCl in the first cycle, followed by two cycles of cold water washing (T3), can be a simple method for increasing gel-forming capability and oxidative stability. The mackerel surimi produced using this washing approach has a higher quality than that produced with regular washing. This straightforward method will enable the sustainable use of dark-fleshed fish for the production of surimi.

**Keywords:** washing; mackerel; fish; surimi; gel; protein content

## 1. Introduction

Fishery and aquaculture production varies according to species, manufacturing, and product type and is mostly used for food. In 2018, around 88% of the total global fish

production of 179 million mt was used as human food, while the remaining 12% was employed for non-food reasons [1]. The use and processing of fish varies greatly across regions. A significant majority of fish output in Asia is sold live or fresh to consumers, whereas fish production in Europe and North America is predominantly offered frozen or preserved [1].

Surimi is one of the fishery products that is intensively manufactured in Asian countries for further processing [2]. Surimi is a refined sort of fish flesh with special techno-functionalities, such as the capacity to generate gels and bind water and oils, making it an important ingredient in a wide range of processed foods [3,4]. Surimi processing technology includes washing minced fish to clean and concentrate the myofibrillar proteins, which are then transformed into additional commodities or stabilized by the incorporation of cryoprotectants, frozen, and kept for use afterward [4]. Optimizing the techno-functionality and quality of surimi generated from raw materials, which are typically under-used, low-value fish species, is a general objective of surimi research studies and production operations [5–7]. Surimi may possibly be made from any fish, but the gelling features of the surimi are determined by the nature of myofibrillar proteins, which are impacted by the species and freshness of the fish, in addition to the processing conditions, primarily in terms of protein concentration, ionic strength, pH, and temperature [5]. Threadfin bream (*Nemipterus* spp.), bigeye snapper (*Priacanthus* spp.), croaker (*Pennahia* and *Johnius* spp.), and lizardfish (*Saurida* spp.) are the principal white fish employed in Southeast Asia for the manufacturing of surimi [2]. The supply of raw materials, however, is the fundamental issue facing the Asian surimi industry [8]. Dark muscle fish have garnered more interest as a potential replacement raw material for the manufacture of surimi owing to the restricted white muscle fish supplies and the excessive harvesting of white fish in Thailand [8]. According to reports, the Gulf of Thailand has seen a rise in the catch of pelagic fish [5,9]. As a result, one of the most difficult ways to convert fish resources into human consumables, especially protein-gated products, is the usage of this small pelagic fish for the manufacturing of surimi. Chaijan et al. [5] examined the gel-forming capacities of surimi from the dark-fleshed mackerel species *Restrelligar kanagurta*, *Restrelligar brachysoma*, and *Auxis thazard*.

According to Singh et al. [10], dark-fleshed fish species naturally have a high concentration of dark muscle that contains significant amounts of lipids and sarcoplasmic proteins. When compared to light muscle, dark muscle has poorer gelation qualities because it contains more sarcoplasmic proteins and lipids [11]. Fish myofibril protein gels' strength, deformability, and color were negatively impacted by sarcoplasmic proteins [5,11]. According to Chaijan et al. [5], myoglobin predominates among the pigment proteins in the sarcoplasmic fraction and is a factor in the surimi gel's decreased whiteness. According to Ochiai et al. [12], removing as much dark muscle as possible will result in high-quality surimi that has better whiteness and a stronger gel. It is challenging to completely remove the dark meat from fish with dark flesh, though. Therefore, the washing procedure is still required for surimi made from the whole muscle of dark-fleshed fish in order to improve color and gel strength.

Washing is the most critical stage in surimi manufacturing to achieve optimal gelling along with colorless and odorless surimi [13]. When minced meat is washed, most of the issues related to color, flavor, and odor are reduced or removed. Approximately two-thirds of minced fish meat is myofibrillar proteins, which are the primary protein in the development of a gel structure. The other one-third is made up of blood, myoglobin, lipids, and sarcoplasmic proteins, which impair the surimi gels' final quality. Consequently, washing, which removes the undesirable one-third, improves the surimi quality by concentrating the myofibrillar protein and extends the frozen storage life [11,13]. Washing involves combining minced fish with cold water and eliminating water using screening, dehydrators, or centrifugation to around 5–10% solids. This procedure is carried out two or three times [14].

Surimi's color can be brightened by extending the washing cycle, washing time, and volume of washing medium [15]. Chen et al. [16] proposed that leaching mince with

ozonized water during a short washing period can improve the color of dark-fleshed fish surimi, such as horse mackerel. However, lipid and protein oxidation can limit its applicability. Somjid et al. [17] developed a new method for obtaining gel-forming surimi from dark muscle fish by employing cold carbonated water (CW) as a washing medium for tropical mackerel (*A. thazard*) surimi manufacturing. The most efficient method for increasing the total quality of mackerel surimi gel was a one-cycle washing with cold CW and then a two-cycle rinsing with cold water [17]. However, myoglobin removal was not enhanced and remains a difficulty. Thus, the objective of this study was to increase the capacity for myoglobin removal from mackerel mince and to enhance the quality of the resulting surimi by incorporating NaCl into the CW washing medium. This assumption was made on the premise that increasing the ionic strength of the washing medium by incorporating NaCl will aid in improving the leaching capacity of myoglobin during washing of mackerel mince, as recently reported by Wang et al. [18]. To reduce myofibrillar protein loss and yield reduction, the concentration of NaCl to be included with CW, as well as the cycle of washing with NaCl, should be tuned. The biochemical features, myoglobin and associated species contents, lipid content, and oxidative stability, as well as gelling characteristics of the resulting surimi, were extensively evaluated in order to finalize a suitable washing solution for straightforward mackerel surimi production.

## 2. Materials and Methods

### 2.1. Fish Sample

Mackerel (*Auxis thazard*) weighing 100–120 g was purchased from a local market in Thasala, Nakhon Si Thammarat, Thailand. The fish were taken approximately a twelve-hour period after capture and wrapped in ice with a fish/ice weight ratio of 1:2 (*w/w*) before being transported within 20 min to Walailak University. Following that, the fish were headed, eviscerated, cleaned, filleted, and skinned. The fish flesh was then thoroughly minced with a meat processor (a Panasonic MK-G20MR, Tokyo, Japan, with a 4 mm aperture diameter).

### 2.2. Surimi Production and Analysis

To study the effect of NaCl concentrations in cold CW washing on properties of mackerel surimi, fresh fish mince was washed with cold CW (Chang®, Cosmos Brewery Co., Ltd., Wangnoi, Ayutthaya, Thailand) in the presence of NaCl (0%, 0.3%, 0.6%, and 0.9%, *w/v*) in the 1st washing cycle with a medium/mince ratio of 3:1 (*v/w*). The 2nd and 3rd cycles were performed using cold tap water without or with NaCl at different concentrations (0.3%, 0.6%, and 0.9%, *w/v*). The maximum NaCl concentration was fixed at 0.9% to prevent the loss of myofibrillar proteins. The mixture was gently stirred for 10 min in a cold environment (4 °C), and the rinsed mince then went through a thin layer of nylon screen. With an ultimate moisture level of 80%, hydraulic press dewatering was applied. As controls, conventional washing with cold tap water for 3 cycles and unwashed mince were employed [17]. All samples were mixed thoroughly prior to being placed in an air-blast freezer with 4% (*w/w*) sucrose and 4% (*w/w*) sorbitol. The frozen samples were maintained at −18 °C until they were analyzed. Figure 1 depicts a schematic illustration of surimi production and analysis. Table 1 illustrates the washing media employed in each treatment as well as their yields.

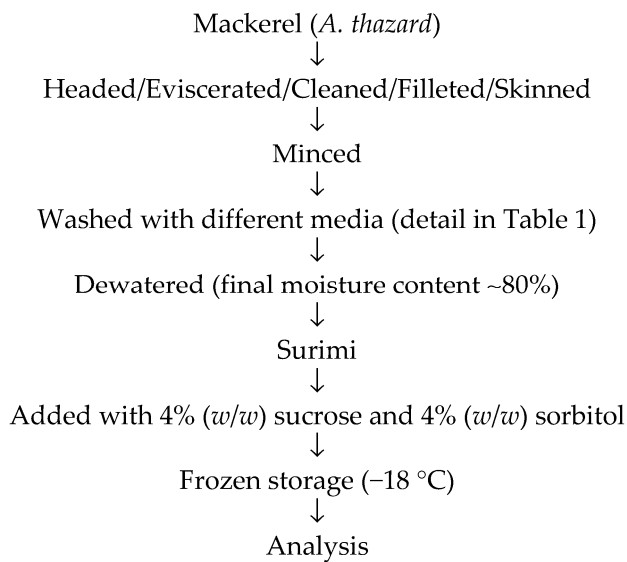

**Figure 1.** A schematic illustration for the surimi production and analysis.

**Table 1.** Washing media used in each treatment and their yields.

| Treatment | 1st Cycle | 2nd Cycle | 3rd Cycle | Yield (%) * |
|---|---|---|---|---|
| Unwashed mince (U) | - | - | - | - |
| T1 | Cold carbonated water (CW) | Cold tap water (CTW) | CTW | 62.27 ± 1.1 a |
| T2 | CW + 0.3% NaCl | CTW | CTW | 54.77 ± 2.3 b |
| T3 | CW + 0.6% NaCl | CTW | CTW | 39.64 ± 2.0 c |
| T4 | CW + 0.9% NaCl | CTW | CTW | 27.24 ± 1.4 f |
| T5 | CW + 0.3% NaCl | CTW + 0.3% NaCl | CTW + 0.3% NaCl | 35.23 ± 2.1 d |
| T6 | CW + 0.6% NaCl | CTW + 0.6% NaCl | CTW + 0.6% NaCl | 30.38 ± 1.0 e |
| T7 | CW + 0.9% NaCl | CTW + 0.9% NaCl | CTW + 0.9% NaCl | 16.69 ± 2.3 g |
| Conventional washing (C) | CTW | CTW | CTW | 39.04 ± 1.9 c |

* Values are given as mean ± standard deviation from 3 determinations. Different letters in the same column represent significant differences ($p < 0.05$).

### 2.2.1. Measurement of Yield, Moisture, and pH

The yield of surimi from all washing procedures was calculated using the weight of the raw fish material in relation to the weight of minced fish. The moisture content and pH of unwashed mince and surimi were determined by the AOAC [19] and Benjakul et al. [20] techniques, respectively.

### 2.2.2. Measurement of Reactive Sulfhydryl (SH), Ca$^{2+}$-ATPase Activity, and Trichloroacetic Acid (TCA) Soluble Peptide

The reactive SH content and Ca$^{2+}$-ATPase activity of natural actomyosin (NAM) isolated from unwashed mince and surimi were measured by the methods of Ellman [21] and Benjakul et al. [20]. TCA-soluble peptide was also measured using the Benjakul et al. [20] technique.

### 2.2.3. Measurement of Protein Hydrophobicity

The hydrophobicity of nonsolubilized myofibrils was measured using bromophenol blue (BPB) for electrophoresis and reported in terms of BPB bound, as highlighted by Chelh et al. [22].

$$\text{BPB bound (μg)} = 200 \text{ μg} \times \left[ \frac{A_{control} - A_{sample}}{A_{control}} \right] \quad (1)$$

where A = absorbance at 595 nm.

### 2.2.4. Measurement of Myoglobin, Non-Heme Iron, and Lipid Contents

The concentration of myoglobin was measured spectrophotometrically at 525 nm by Benjakul and Bauer's approach [23]. Schricker et al. [24] devised a method for measuring non-heme iron concentration. The Bligh and Dyer procedure [25] was employed for the extraction of the lipid.

### *2.3. Gel Preparation*

To defrost frozen samples, running tap water was used until the interior temperature approximated 0 °C. Following that, the samples were cut into small fragments, and the moisture level was fixed to 80%. Dry NaCl (2.5% *w/w*) was introduced to the samples and chopped for 5 min to generate a homogeneous sol. Next, the sol was inserted in a polyvinylidene casing (∅ = 2.5 cm), and both sides were tightly sealed. Following a 30 min incubation period at 40 °C, the sample underwent a 20 min heating period at 90 °C. Consequently, the gels were chilled in ice water before being stored at 4 °C for 24 h [7].

### *2.4. Gel Analyses*

The gel strength (breaking force × deformation) of unwashed mince and surimi gels was reported after the texture was evaluated using a texture analyzer (TA-XT2, Stable Micro Systems, Godalming, Surrey, UK) mounted with a spherical plunger (diameter 5 mm; depression speed 60 mm/min/trigger force 0.05 N) [7]. The expressible moisture of gels was calculated as a percentage of sample mass using the procedure of Phetsang et al. [7]. The values of *L\**, *a\**, and *b\** were recorded with a colorimeter (Hunter Assoc. Laboratory; Reston, VA, USA), and the whiteness was computed as follows [7]:

$$\text{Whiteness} = 100 - [(100 - L^*)^2 + a^{*2} + b^{*2}]^{1/2} \quad (2)$$

The thiobarbituric acid reactive substance (TBARS) assay developed by Buege and Aust [26] was performed using the ground sample.

A scanning electron microscope (SEM) (GeminiSEM, Carl Zeiss Microscopy, Oberkochen, Germany) operating at a 10 kV accelerating voltage was used to examine the microstructures of gels [27].

Ten trained panelists with extensive experience evaluating off-flavors of seafood evaluated the gel. The severity of fishy odor was assessed on a 5-point scale ranging from none to intense (score 0–4) [8]. The Walailak University Human Research Ethics Committee authorized the study procedure (WUEC-21-125-02).

### *2.5. Statistical Analysis*

Throughout the study, a completely randomized design was employed. ANOVA was utilized to manage the data, and mean comparisons were made using Duncan's Multiple Range Test. For the statistical analysis, the statistical program (SPSS 23.0, SPSS Inc., Chicago, IL, USA) was employed.

## 3. Results and Discussion

### *3.1. Yield*

Table 1 shows the yield of surimi made with various washing mediums. It has been reported that around 50% of total protein disappears during washing, but this

can vary depending on the washing conditions [28]. The maximum yield was obtained by washing with CW in the first cycle and water in the second and third cycles (T1) ($p < 0.05$). When varying quantities of NaCl were mixed into the first washing medium (T2–T4), the yield decreased with rising NaCl content ($p < 0.05$). When washing with NaCl for three cycles (T5–T7), the yield was greatly decreased ($p < 0.05$). In this case, the drop in yield was caused by the leaching of myofibrillar proteins with increasing ionic strength, particularly with extensive washing by elevating the round of washings in the presence of NaCl. Conventional surimi washing (C) produced a yield of roughly 40%, which was comparable to T3. Too much salt in the wash water has been observed to promote the solubilization of myofibrillar proteins, leading to a higher loss of myofibrillar proteins and lower yield [29,30]. As NaCl rose by more than 1.0% in the washing medium, it reduced the protein recovery of grass carp surimi [18]. It has been discovered that washing grass carp with 0.25–0.5% NaCl at pH 5–7 can significantly improve the yield of surimi [18]. The application of NaCl in the washing solution was chosen because NaCl, one of the least expensive substances, is a critical component and a processing aid for a variety of food products [31]. Other NaCl alternatives can be researched in the future to increase the yield and quality of mackerel surimi. However, the proposed compounds' cost and efficiency should be carefully assessed.

Myofibrillar proteins are the majority of muscle proteins that possess desirable gelling, emulsifying, and film-forming properties [32]. Myofibrillar proteins are thought to be salt-soluble proteins [33]. Muscle tissue's physiological ionic strength is predicted to be 0.15–0.30 M [34]. Additional salts, particularly NaCl, are usually employed during the processing to enhance the flavor, color, texture, and shelf-life of muscle-based foods. To solubilize the myofibrillar proteins and attain the appropriate functional characteristics, 2–3% (0.47–0.68 M) NaCl is usually required [35]. Although the salt concentration in the washing media was maintained at no more than 0.9% to prevent myofibrillar protein loss, some myofibrillar proteins were still eliminated during washing, especially with higher salt content and the presence of salt in the first and second washing cycles. The DLVO theory, which claims that colloidal stability is determined by a compromise of attractive van der Waals forces and repulsive double-layer electrostatic bonds, can be used to explain the impact of ions on the solubility of the protein in colloidal research [36]. Protein can be thought of as a macroion. Under a salt solution environment, it is enclosed by more counterions than co-ions, which shields the protein surface charge and allows it to solubilize. As the salt content rises, the protective role of protein charge causes the electric double layers to compress and the repulsive term of the DLVO theory to diminish [36]. Protein solubility is projected to reduce at high salt concentrations [37,38].

Based on the results, treatments that produced lower yields than conventional washed surimi (C) were eliminated in order to maximize the use of fish resources and for commercial reasons. The maximum NaCl content in CW can be set at 0.6% only during the first washing cycle (T3).

*3.2. Biochemical Features*

3.2.1. pH

The pH of mackerel mince was 5.54, and the pH of surimi increased significantly, depending on the medium employed (Table 2). The highest pH value of 6.60 was found in T1 surimi ($p < 0.05$). Surimi conventionally washed with water (C), T2, and T3 showed a comparable pH value (pH ~ 6.1) ($p > 0.05$), which was in agreement with Chaijan et al. [39], who reported a pH value of 6.17 in mackerel (*R. kanagurta*) surimi produced with 0.5% NaCl leaching. In the case of CW washing (T1–T3), the pH tended to be lower, with NaCl content in the washing medium, regardless of concentration. It has been reported that the ultimate pH value of surimi may be related to the pH of the medium applied and can be impacted by the effectiveness of leaching along with the adsorption of washing media into surimi. Upon the washing step, both acidic and alkaline elements can be leached out of fish mince, leading to a new muscle pH equilibrium [17].

**Table 2.** Effect of cold carbonated water and NaCl washing on biochemical properties of mackerel surimi in comparison with unwashed mince and conventional surimi.

| Treatment | pH | Reactive Sulfhydryl Content (mol/$10^8$ g Protein) | $Ca^{2+}$-ATPase Activity (µmolPi/mg Protein/min) | Surface Hydrophobicity; BPB Bound (µg) | TCA-Soluble Peptide (µmol Tyrosine/g Sample) |
|---|---|---|---|---|---|
| U | 5.54 ± 0.01 c | 2.70 ± 0.36 bc | 0.47 ± 0.02 a | 34.09 ± 0.50 e | 0.23 ± 0.00 a |
| T1 | 6.60 ± 0.22 a | 3.45 ± 0.24 ab | 0.14 ± 0.02 e | 52.66 ± 3.51 c | 0.12 ± 0.01 b |
| T2 | 6.11 ± 0.14 b | 2.19 ± 0.06 c | 0.29 ± 0.04 b | 43.17 ± 3.46 d | 0.12 ± 0.00 b |
| T3 | 6.06 ± 0.04 b | 3.61 ± 0.58 a | 0.20 ± 0.03 cd | 64.02 ± 3.78 b | 0.11 ± 0.01 b |
| C | 6.05 ± 0.06 b | 3.46 ± 0.34 ab | 0.24 ± 0.01 bc | 76.27 ± 3.18 a | 0.11 ± 0.01 b |

Values are given as mean ± standard deviation from three determinations. Different letters in the same column indicate significant differences ($p < 0.05$). BPB = bromophenol blue. TCA = trichloroacetic acid. U = unwashed minced. T1 = cold carbonated water washing in the 1st washing cycle and cold tap water washing in the 2nd and 3rd cycles. T2 = cold carbonated water washing in the presence of 0.3% NaCl (*w/v*) in the 1st washing cycle and cold tap water washing in the 2nd and 3rd cycles. T3 = cold carbonated water washing in the presence of 0.6% NaCl (*w/v*) in the 1st washing cycle and cold tap water washing in the 2nd and 3rd cycles. C = conventional washing with 3 cycles of cold tap water.

### 3.2.2. Reactive SH, $Ca^{2+}$-ATPase Activity, and Surface Hydrophobicity

Biochemical changes, such as reactive SH, $Ca^{2+}$-ATPase activity, and surface hydrophobicity in this case, can show that the structure of fish proteins has changed during the manufacturing of surimi [27]. Those indices were altered in various ways depending on the washing approach (Table 2). Myosin molecules' head and tail segments frequently contained a number of reactive SH groups, according to Buttkus [40]. When the molecular structure of myosin changes, the reactive SH groups may become more exposed [20]. After washing, surimi's reactive SH concentration changed ($p < 0.05$; Table 2). Therefore, washing can accelerate the protein denaturation, exposing hidden SH and accelerating the oxidation of SH groups. Unwashed mince and T2 surimi had the next-highest SH levels, followed by T1, T3, and conventional surimi.

Unwashed mince exhibited the highest $Ca^{2+}$-ATPase activity ($p < 0.05$; Table 2). Regardless of the washing media, a decline in the $Ca^{2+}$-ATPase activity was seen in all surimi ($p < 0.05$). In comparison with unwashed mince, Das et al. [41] and Somjid et al. [27] observed that the $Ca^{2+}$-ATPase activities in mackerel, pink perch, croaker, and sardine surimi gradually decreased. A reduction in NAM's $Ca^{2+}$-ATPase activity suggested that myosin integrity was lost due to denaturation and/or aggregation [42]. The structural alterations of the myofibrillar proteins upon washing are responsible for the loss of $Ca^{2+}$-ATPase activity [41]. Unwashed mince and surimi have $Ca^{2+}$-ATPase activity in the following order: U > T2 ≥ C ≥ T3 > T1.

Table 2 also shows the surface hydrophobicity of surimi and unwashed mince. Since hydrophobic amino acid sequences are usually found inside protein structures, the surface hydrophobicity of protein molecules is typically related to their exposed hydrophobic moieties [43]. Exposed hydrophobic residues can create contacts and enhance the gel structure during thermal gelation [39,44]. A good gel-forming capability of protein, on the other hand, requires a sufficient shift in surface hydrophobicity [17]. According to the findings, unwashed mince carried the lowest surface hydrophobicity ($p < 0.05$). Surimi with the greatest surface hydrophobicity ($p < 0.05$) was conventional surimi, followed by T3, T1, and T2. Among the CW washes, the one with 0.6% NaCl (T3) produced surimi with higher surface hydrophobicity than the others. Wang et al. [18] discovered that increasing the NaCl concentration during washing elevated the α-helix content of actomyosin, an indication of protein unfolding.

### 3.2.3. TCA-Soluble Peptide

Table 2 displays the TCA-soluble peptide contents of unwashed mackerel mince, typical surimi, and surimi washed with CW-NaCl solutions. The TCA-soluble peptide

in unwashed mince was 0.23 μmol tyrosine/g sample, and the value was reduced by approximately 50% following washing, regardless of washing medium. TCA-soluble peptide indicated proteolytic breakdown and a higher concentration indicated greater muscle protein hydrolysis [20]. According to Somjid et al. [27], unwashed mince provided the most TCA-soluble peptide, suggesting the presence of some proteinases that can produce proteolytic breakdown products. During handling and storage, endogenous proteinases in the sarcoplasmic fraction can trigger autolysis and buildup of particular soluble peptides in mackerel mince [27]. By repeatedly washing the mince, peptides can be removed, and proteinases may be inactivated. Due to proteolytic enzymes' association with myofibrillar proteins and interference with gelation, dark-fleshed fish species have weak gelation ability [45]. Activation of myofibril-bound proteinases during the early phases of thermal gelation can result in gel-weakening [45].

### 3.3. Myoglobin, Non-Heme Iron, and Lipid Contents

Unwashed mince contained about 27 mg/100 g of myoglobin. As a result, regardless of the washing method, the myoglobin content of surimi was lower than that of unwashed mince ($p < 0.05$) (Figure 2a). The results revealed that washing with cold tap water or CW could remove some myoglobin. However, due to the association among myoglobin and muscle components, particularly myofibrillar proteins, it is difficult to separate all myoglobin from mackerel mince [5]. When washed with the conventional process (C), the myoglobin content decreased by 46.76%. However, CW washing (T1) only removed 3.8% of the myoglobin. This was consistent with a previous study in which CW washing enhanced the gel functionality of mackerel surimi but did not increase heme protein removal [17]. When salt was added to the CW medium, the myoglobin removal efficacy increased to 36.68% for T2 and 64.63% for T3. The presence of salt at the optimal level may have aided in weakening the association between myoglobin and myofibrillar proteins or muscle components, hence facilitating myoglobin leaching efficacy. As previously indicated, raising the ionic strength of the washing medium can improve the solubility of myofibrillar proteins. This can cause myoglobin to dissociate from the myoglobin–myofibrillar complex, allowing myoglobin to be eliminated within the washing water. Unwashed mince had a non-heme iron level of 50.63 mg/g, which significantly decreased following washing with all available methods (Figure 2a). Non-heme iron was eliminated by 93.72% by T3, compared to 79.68%, 68.56%, and 65.83% by T2, T1, and C, respectively. T3 may remove the non-heme iron through washing media due to the enhanced solubility of myofibrillar proteins at high ionic strength. Overall, the removal of myoglobin and non-heme iron from mackerel by T3 appeared to be the most successful washing procedure, which may have enhanced the gel's functionality and oxidative stability.

Figure 2b depicts the lipid content of unwashed mince and surimi obtained by conventional washing and CW-NaCl washing. Lipid is regarded as an unfavorable surimi component because it interferes with gel formation and induces gel rancidity [39,46]. Unwashed mince included about 2 g/100 g lipid, which was removed to variable degrees after washing depending on the washing solution employed. According to the previous work [17], conventional washing eliminated nearly half of the lipid content in fish mince. CW washing (T1) removed 52% of the lipid, and the removal capacity improved to 63 and 80% when washed with T2 and T3, respectively, as the NaCl level in the washing medium increased. This was most likely due to the solubility of myofibrillar proteins, which can aid in the removal of lipids from muscle components. The reduced yield of surimi produced by T3 (Table 1) may potentially be related to the residual lipid content. However, it has been noted that oxidative stability and gel-forming ability increase with decreasing lipid concentration [46].

**(a)**

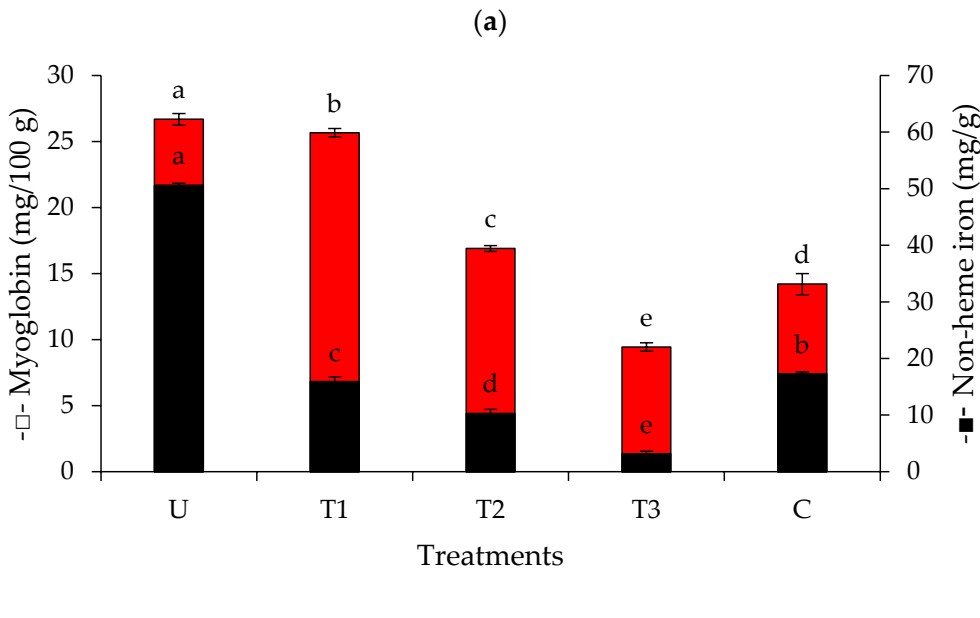

**(b)**

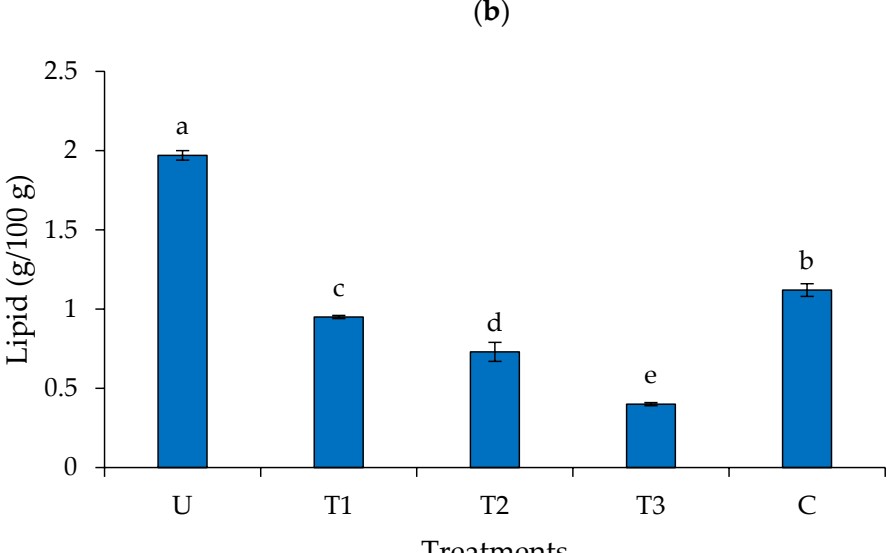

**Figure 2.** Effect of cold carbonated water and NaCl washing on myoglobin and non-heme iron contents (**a**) and lipid content (**b**) of mackerel surimi in comparison with unwashed mince and conventional surimi. Bars indicate standard deviation from 3 determinations. Different letters on the bars represent significant differences ($p < 0.05$). U = unwashed minced. T1 = cold carbonated water washing in the 1st washing cycle and cold tap water washing in the 2nd and 3rd cycles. T2 = cold carbonated water washing in the presence of 0.3% NaCl ($w/v$) in the 1st washing cycle and cold tap water washing in the 2nd and 3rd cycles. T3 = cold carbonated water washing in the presence of 0.6% NaCl ($w/v$) in the 1st washing cycle and cold tap water washing in the 2nd and 3rd cycles. C = conventional washing with 3 cycles of cold tap water.

*3.4. Gelling Properties*

3.4.1. Gel Strength, Expressible Drip, and Whiteness

Unwashed mince consisted of more lipid, myoglobin, non-heme iron, and TCA-soluble peptide, all of which had a negative impact on the ability of myofibrillar proteins to form gels. From the unwashed minced gel, the lowest gel strength (Figure 3a), most expressible drip (Figure 3b), and lowest whiteness (Figure 3c) can be obtained. Furthermore, one of the elements influencing the textural characteristics of surimi gel may be its pH. Unwashed mince had a pH of 5.54, which was close to the isoelectric point of myosin (pI 5.0–5.5), which

is more likely to promote agglutination with water released from the matrix than other treatments with higher pH values (pH 6–7) (Table 2). Surimi washed with all treatments performed better in terms of gel strength, water-holding capacity, and whiteness than unwashed mince (Figure 3). T3 produced the highest gel strength surimi, followed by T1/T2 and conventional surimi. T3 may increase the ionic strength in the washing medium to achieve the best conditions for protein dissociation and denaturation in the subsequent step, hence increasing gel strength. Among CW washing, surimi rinsed with T3 tended to exhibit the highest reactive SH content and surface hydrophobicity (Table 2), indicating that those groups can join to create a three-dimensional network with water-binding potential. Surimi's SH groups may encourage the development of disulfide bonds, which are essential for gel strengthening after gelation [47]. The T2 with the lowest SH group among the surimi may create a gel with a higher water release. The conventional surimi had the maximum hydrophobicity (Table 2), which may prevent the appropriate protein–protein interaction from forming a gel. Protein unfolding has been shown to expose hydrophobic residues, altering the surface hydrophobicity of the protein [48]. Exposed hydrophobic residues can create contacts and strengthen the gel network during thermal gelation [39,44]. A good gel-forming property of protein, on the other hand, requires a sufficient shift in surface hydrophobicity [17].

Surimi gel had a higher level of whiteness than unwashed mince, and among the surimi, the gel from T3 had the highest whiteness. Surimi's color was influenced by its residual pigment content and thermal gelation stability. Furthermore, components like lipids, which can oxidize when heated, can serve as an additional source of color via the Maillard reaction because lipid oxidation produces aldehyde, which can then combine with amine to create brown pigment via the amine–carbonyl reaction or Maillard reaction [27,49]. T3's gel was the whitest because it had the least amount of myoglobin, non-heme iron, and lipid (Figure 2), as well as the lowest degree of lipid oxidation (see below).

In the investigation of another species of mackerel, *Scomber japonicus*, washing was performed using 4 vol of 0.04 M phosphate buffer (pH 8.0 and 8.5) or 0.04 M bicarbonate buffer (pH 8.0 and 8.5) at 4 °C for 5 min in the first cycle, followed by two cycles of 3 vol of chilled tap water (4 °C). During the final washing, the pH of the mince was decreased to 7.0 by using a 1.0 M citric acid solution. The moisture content of washed mince was reduced to 78% by centrifugation for 15–20 min at $1500 \times g$ [50]. Washing with alkaline phosphate or bicarbonate buffers did not remove the pigment from minced mackerel [50]. Because the alkaline treatment alone was ineffective at improving the color of mackerel surimi, ozonation was attempted. Ozonation at pH 3.0 for 30 min improved the color of mackerel surimi. To improve the gel strength, the ozonated mackerel mince could be blended with 0.1% $NaHSO_3$, 0.2% ascorbic, 0.15% cysteine, or their combination after regaining the pH to neutral on the final washing [50].

According to the findings of those studies, a complex step was used in the production of mackerel surimi [50], alkaline washing did not improve the color of dark-fleshed fish, and the use of ozone, an oxidizing agent, may have a negative impact on gel-forming ability and oxidative stability of lipid in the resulting surimi. Thus, washing with cold carbonated water containing 0.6% NaCl in the first washing cycle, followed by two cycles of cold tap water washing (T3), can be a straightforward method for producing good surimi from mackerel mince.

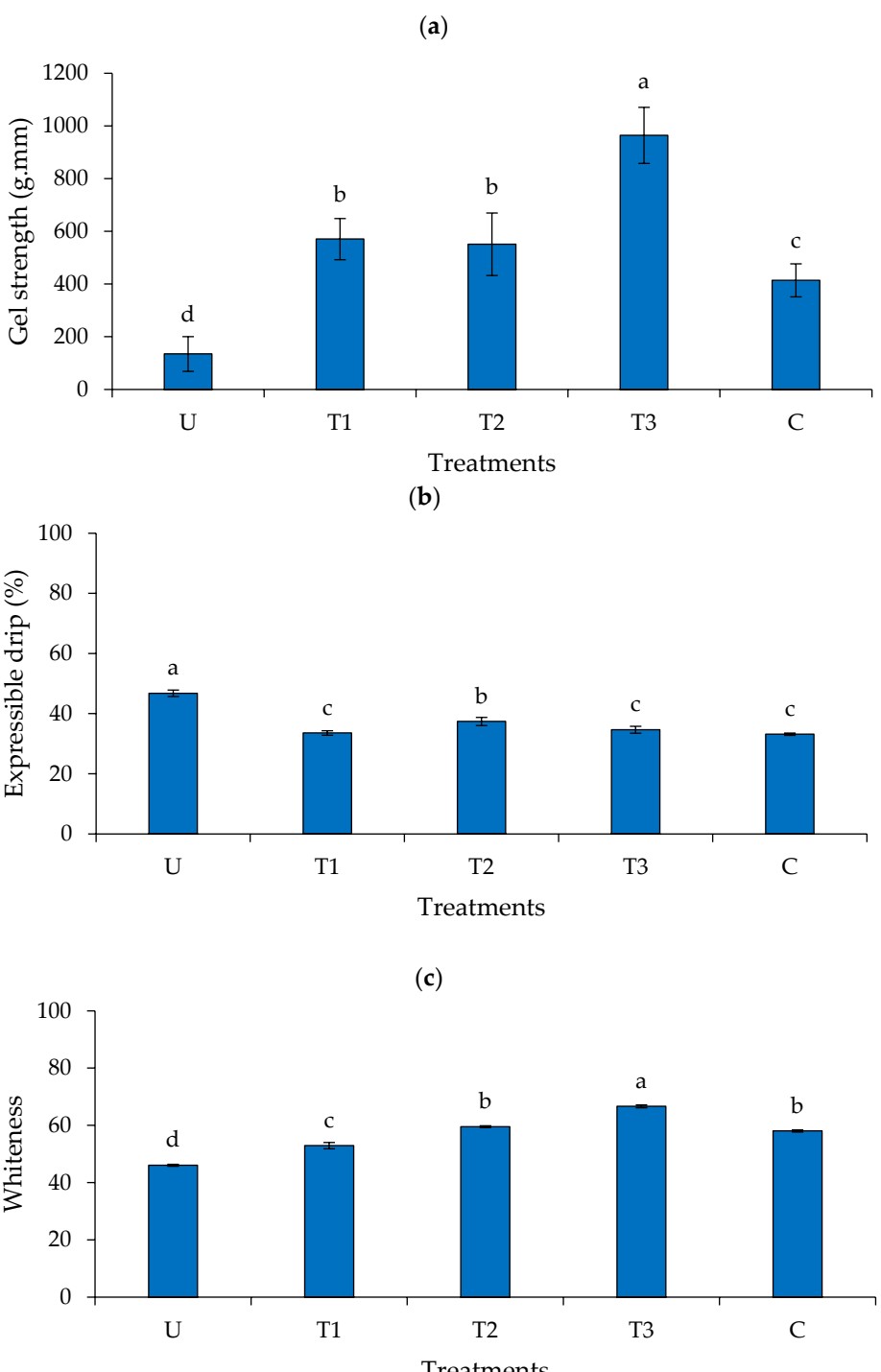

**Figure 3.** Effect of cold carbonated water and NaCl washing on gel strength (**a**), expressible drip (**b**), and whiteness (**c**) of mackerel surimi gel in comparison with gels from unwashed mince and conventional surimi. Bars indicate standard deviation from 3 determinations. Different letters on the bars represent significant differences ($p < 0.05$). U = unwashed minced. T1 = cold carbonated water washing in the 1st washing cycle and cold tap water washing in the 2nd and 3rd cycles. T2 = cold carbonated water washing in the presence of 0.3% NaCl ($w/v$) in the 1st washing cycle and cold tap water washing in the 2nd and 3rd cycles. T3 = cold carbonated water washing in the presence of 0.6% NaCl ($w/v$) in the 1st washing cycle and cold tap water washing in the 2nd and 3rd cycles. C = conventional washing with 3 cycles of cold tap water.

### 3.4.2. Microstructure

Surimi gel's fine network structure was visible in T1, T2, and, notably, T3 (Figure 4). Overall, T1, T2, and T3 surimi gel showed routinely continuous and smooth network architectures due to the removal of non-gel-forming components, including lipid and myoglobin [8] with appropriate unfolding [17,39,44], which might be linked to the proper gel network formation of myofibrillar proteins. During thermal gelation, myosin and other salt-soluble myofibrillar proteins undergo complicated changes in rheological features. A good surimi gel can be generated by constructing a regular assembled structure with a well-organized three-dimensional matrix [8,17]. The network of the conventional surimi (C) gel had some holes in it and a discontinuous structure, which may have contributed to its reduced gel strength compared to the surimi gel washed with CW-NaCl (T1–T3) (Figure 4). Unwashed mince had a protein network packed with clots and aggregates, which was associated with a weaker gel and more water expelled (Figure 3). Sarcoplasmic proteins, according to Arfat and Benjakul [51], can interact with myofibrillar proteins and impede the formation of a robust gel structure. Residual lipids were reported to impair the gel-forming capacity of surimi because lipids are unable to gel and tend to interfere with the protein networks [8]. Overall, the finest structure of T3 surimi gel was associated with the highest gel strength (Figure 3a). As a result, T3 was determined to be the optimal washing method for producing good gel-forming surimi from tropical mackerel.

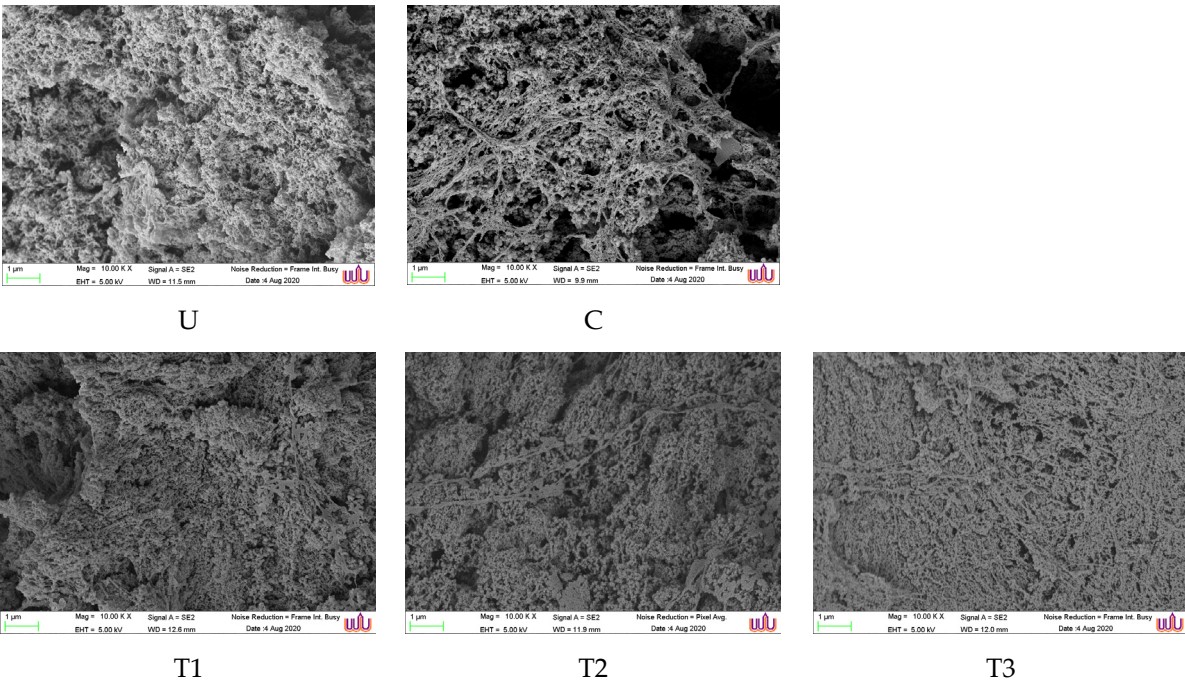

**Figure 4.** Microstructure of gels from unwashed mince and surimi from mackerel surimi. (Magnification: 10,000× EHT: 5.0 kV). U = unwashed minced. T1 = cold carbonated water washing in the 1st washing cycle and cold tap water washing in the 2nd and 3rd cycles. T2 = cold carbonated water washing in the presence of 0.3% NaCl (*w/v*) in the 1st washing cycle and cold tap water washing in the 2nd and 3rd cycles. T3 = cold carbonated water washing in the presence of 0.6% NaCl (*w/v*) in the 1st washing cycle and cold tap water washing in the 2nd and 3rd cycles. C = conventional washing with 3 cycles of cold tap water.

### 3.4.3. Lipid Oxidation and Fishy Odor

Lipid oxidation of gels from unwashed mince and surimi washed with different media, as monitored by the TBARS assay, is shown in Figure 5a. Generally, all the surimi gels showed decreased TBARS levels than unwashed mince gel. This was due to the elimination of lipid and pro-oxidants like myoglobin and non-heme iron during washing (Figure 2).

Among the surimi, conventional surimi gel had the highest TBARS content, followed by T1/T2 and T3, respectively. The lowest TBARS value in T3 surimi gel was also linked to the lowest residual lipid content as well as myoglobin and non-heme iron contents (Figure 2).

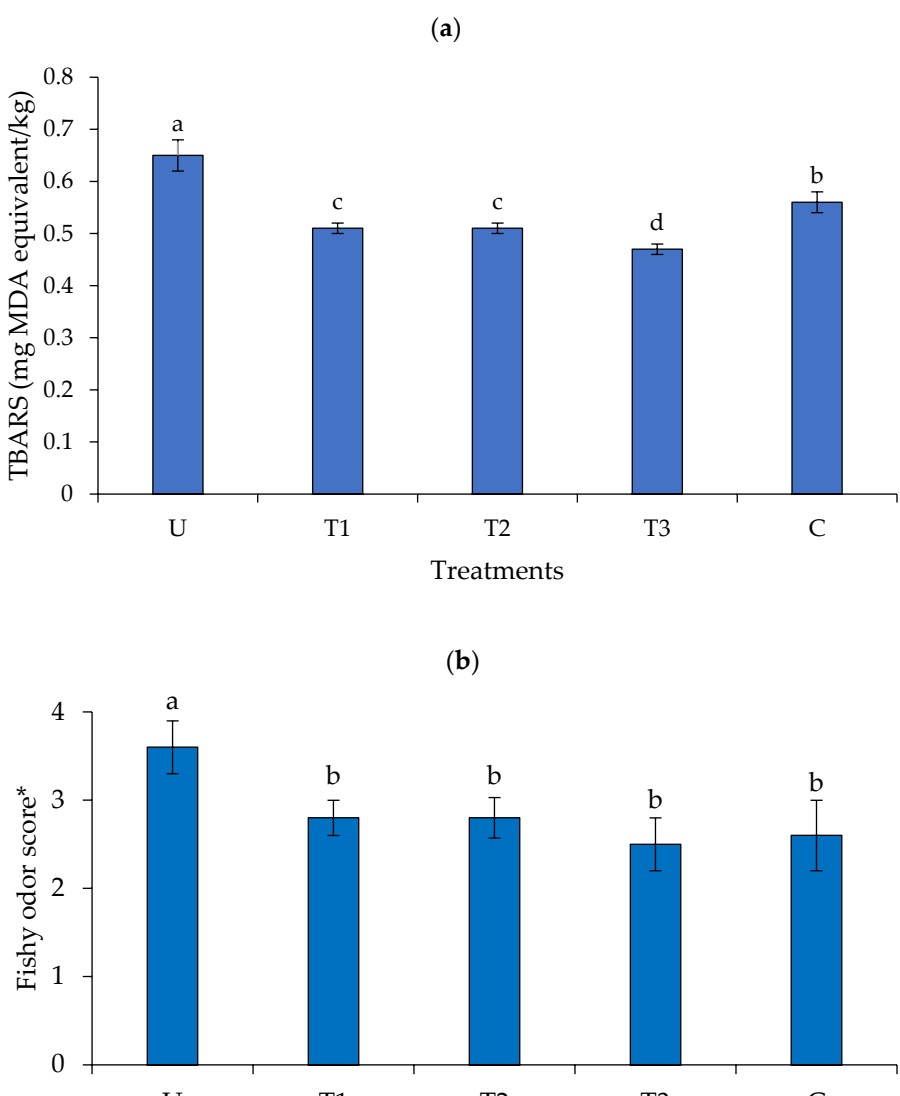

**Figure 5.** Effect of cold carbonated water and NaCl washing on lipid oxidation measured as TBARS assay (**a**) and fishy odor score (**b**) of mackerel surimi gel in comparison with gels from unwashed mince and conventional surimi. Bars indicate standard deviation from 3 determinations except for fishy odor score ($n = 10$). Different letters on the bars represent significant differences ($p < 0.05$). U = unwashed minced. T1 = cold carbonated water washing in the 1st washing cycle and cold tap water washing in the 2nd and 3rd cycles. T2 = cold carbonated water washing in the presence of 0.3% NaCl ($w/v$) in the 1st washing cycle and cold tap water washing in the 2nd and 3rd cycles. T3 = cold carbonated water washing in the presence of 0.6% NaCl ($w/v$) in the 1st washing cycle and cold tap water washing in the 2nd and 3rd cycles. C = conventional washing with 3 cycles of cold tap water. A score of 4 represented 'very strong' while 0 represented 'none'.

The TBARS assay is a measure of aldehydes that can be generated and vaporized during heating, therefore altering the TBARS value [7]. In the current investigation, the TBARS content in all surimi gel was within the allowable threshold (TBARS < 1.5 mg MDA/kg) [52]. Surimi gel, particularly when prepared with T3, seemed to be preserved against lipid peroxidation, as evidenced by the lowest TBARS level (0.47 mg MDA equiv-

alent/kg). Overall, T3 washing was found to successfully postpone the formation of lipid oxidation.

Surimi received a lower fishy odor score (2.5–2.8 out of 4) from all panelists than unwashed mince, with no significant variations between treatments ($p > 0.05$). Unwashed mince received the highest fishy odor score, whereas surimi was rated the lowest due to the removal of volatile molecules, pigments, and other odorous substances in fish muscle, which was adjusted by washing solution [27]. Some compounds may be identified postmortem endogenously, while others may be created during the handling and washing stages. The washing media utilized in this investigation had no effect on the fishy odor of the resultant surimi.

## 4. Conclusions

Washing with CW containing 0.6% NaCl in the first washing cycle, followed by two cycles of cold water washing, can be a simple technique to increase the gelling capability and oxidative retention of mackerel surimi. This medium effectively removes lipid, myoglobin, non-heme iron, and TCA-soluble peptide contents, which reduces lipid oxidation while also enhancing gel-forming ability in dark-fleshed fish, owing to the induction of optimal unfolding as reported by some biochemical features such as reactive SH content, $Ca^{2+}$-ATPase activity, and hydrophobicity. The outcomes of this study will help to establish the practicality of tropical mackerel, a dark-fleshed fish species, as a resource for surimi production, which will establish the sustainability of the surimi industry. In the future, upscale production of mackerel surimi using the proposed solution can be performed to advance its potential application in industry. It is also possible to investigate the impact of frozen storage on the gel-forming ability and oxidative stability of the produced surimi.

**Author Contributions:** Conceptualization, W.P. and M.C.; methodology, W.P., H.K.Ç. and M.C.; software, W.P. and M.C.; validation, W.P., H.K.Ç. and M.C.; investigation, P.T., S.B., W.P. and M.C.; resources, W.P. and M.C.; data curation, P.T., S.B., W.P. and M.C.; writing—original draft preparation, W.P. and M.C.; writing—review and editing, W.P., M.C. and H.K.Ç.; funding acquisition, W.P. and M.C. All authors have read and agreed to the published version of the manuscript.

**Funding:** This project was funded by the National Research Council of Thailand (NRCT) (NRCT5-RSA63019-01).

**Institutional Review Board Statement:** The study was conducted according to the guidelines of the Declaration of Helsinki and approved by the Ethics Committee of Walailak University (WUEC-21-125-02; date of approval 8 June 2022).

**Informed Consent Statement:** Informed consent was obtained from all subjects involved in the study.

**Data Availability Statement:** Data are contained within the article.

**Acknowledgments:** The authors thank the Food Technology and Innovation Research Center of Excellence, Walailak University, for facility support.

**Conflicts of Interest:** The authors declare no conflict of interest.

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
