# Peer review of "Surimi Production from Tropical Mackerel: A Simple Washing Strategy for Better Utilization of Dark-Fleshed Fish Resources"

_resources, doi:10.3390/resources12100126_

Round 1

Reviewer 1 Report

This work reported a simple washing strategy for better utilization of dark-fleshed fish resource. The results shown that washing mackerel surimi (A. thazard) with CW containing 0.6% (w/v) NaCl in the 1st cycle, followed by two cycles of cold water washing (T3), can be a simple method for increasing gel-forming ability and oxidative stability. It is well planned, the results are properly described and discussed, and the conclusions are sound and supported by the data. Thus, I recommend this manuscript for publication in Resources after some revisions towards the following points:

1.     The authors should rephrase the abstract such that it better reflects the contents.

2.     Some Figure descriptions need to be further enhanced, such as Figure 2 and Figure 3.

3.     Overall the discussion is ok but could be improved, especially the analysis of the microstructure.

4.     Compared with other reported method, what are the advantages of the simple washing strategy?

5.     For the references, it is recommended that the author reduce to about 45.

Author Response

Reviewer#1

This work reported a simple washing strategy for better utilization of dark-fleshed fish resource. The results shown that washing mackerel surimi (A. thazard) with CW containing 0.6% (w/v) NaCl in the 1st cycle, followed by two cycles of cold water washing (T3), can be a simple method for increasing gel-forming ability and oxidative stability. It is well planned, the results are properly described and discussed, and the conclusions are sound and supported by the data. Thus, I recommend this manuscript for publication in Resources after some revisions towards the following points:

  1. The authors should rephrase the abstract such that it better reflects the contents.

Ans: The abstract has been modified with quantitative data to support the statement. The numerical findings were presented to provide more information and to highlight key issues. The conclusion was also included in the Abstract.

  1. Some Figure descriptions need to be further enhanced, such as Figure 2 and Figure 3.

Ans: All of the figure legends and table captions were modified to clarify the treatments.

  1. Overall the discussion is ok but could be improved, especially the analysis of the microstructure.

Ans: Thank you very much. Additional details about microstructure was provided.

Surimi gel's fine network structure was visible in T1, T2, and notably T3 (Fig. 3). Overall, T1, T2, and T3 surimi gel showed routinely continuous and smooth network architectures due to the removal of non-gel-forming components including lipid and myoglobin [8] with appropriate unfolding [17, 39, 44], which might be linked to the proper gel network formation of myofibrillar proteins. During thermal gelation, myosin and other salt-soluble myofibrillar proteins undergo complicated changes in rheological features. A good surimi gel can be generated by constructing a regular assembled structure with a well-organized three-dimensional matrix [8, 17]. The network of the conventional surimi (C) gel had some holes in it and a discontinuous structure, which may have contributed to its reduced gel strength compared to the surimi gel washed with CW-NaCl (T1-T3) (Fig. 3). Unwashed mince had a protein network packed with clots and aggregates, which was associated with a weaker gel and more water expelled (Fig. 2). Sarcoplasmic proteins, according to Arfat and Benjakul [50], can form interaction with myofibrillar proteins and impede the formation of a robust gel structure. Residual lipids were reported to impair the gel-forming capacity of surimi. Because lipids are unable to gel and tends to interfere the protein networks [8]. Overall, the finest structure of T3 surimi gel was associated with the highest gel strength (Fig. 2a). As a result, T3 was determined to be the optimal washing method for producing a good gel-forming surimi from tropical mackerel.”

  1. Compared with other reported method, what are the advantages of the simple washing strategy?

Ans: A simple washing approach, as shown in the discussion, can remove more myoglobin, non-heme iron, and lipid from mackerel muscle than conventional water washing. However, a comparison with alkaline washing and ozonation was made. A reference was also updated.

Jiang, S.T.; Ho, M.L.; Jiang, S.H.; Lo, L.; Chen, H.C. Color and quality of mackerel surimi as affected by alkaline washing and ozonation. J. Food Sci. 1998, 63(4), 652-655.

In the investigation of other species of mackerel, Scomber japonicus, washing was done using 4 vol of 0.04 M phosphate buffer (pH 8.0 and 8.5) or 0.04 M bicarbonate buffer (pH 8.0 and 8.5) at 4 °C for 5 min in the first cycle, followed by 2-cycles of 3 vol of chilled tap water (4 °C). During the final washing, the pH of the mince was decreased to 7.0 by using a 1.0 M citric acid solution. The moisture content of washed mince was reduced to 78% by centrifugation for 15-20 min at 1500 ×g [50]. Washing with alkaline phosphate or bicarbonate buffers did not remove the pigment from minced mackerel [50]. Because the alkaline treatment alone was ineffective at improving the color of mackerel surimi, ozonation was tried. Ozonation at pH 3.0 for 30 min improved the color of mackerel surimi. To improve the gel strength, the ozonated mackerel mince could be blended with 0.1% NaHSO3, 0.2% ascorbic, 0.15% cysteine, or their combination after regaining the pH to neutral on the final washing [50].

According to the findings of those studies, a complex step was used in the production of mackerel surimi [50], alkaline washing did not improve the color of dark-fleshed fish, and the use of ozone, an oxidizing agent, may have a negative impact on gel-forming ability and oxidative stability of lipid in the resulting surimi. Thus, washing with cold carbonated water containing 0.6% NaCl in the first washing cycle, followed by two cycles of cold tap water washing (T3), can be a straightforward method for producing good surimi from mackerel mince.

.

  1. For the references, it is recommended that the author reduce to about 45.

Ans: To complete the work, we would like to keep the references to support the background, introduction, methods, and discussion.

Reviewer 2 Report

The article titled "Surimi Production from Tropical Mackerel: A Simple Washing Strategy for Better Utilization of Dark-Fleshed Fish Resource" is suitable for publication. However, the quality of presentation of the results is not adequate. Table 2, for example, does not present a description of the treatments in the caption. The black and white figures do not highlight the results, some letters are very close to the error bar (deviations) and for the figure to be clear there must be a description of the treatments in the caption so that it is not necessary to go back to Table 1 to see what it is. It is also necessary to improve the discussion of results, especially in the microstructure part of the gels. With the suggested corrections, the article can be published.

Author Response

Reviewer#2

The article titled "Surimi Production from Tropical Mackerel: A Simple Washing Strategy for Better Utilization of Dark-Fleshed Fish Resource" is suitable for publication. However, the quality of presentation of the results is not adequate. Table 2, for example, does not present a description of the treatments in the caption. The black and white figures do not highlight the results, some letters are very close to the error bar (deviations) and for the figure to be clear there must be a description of the treatments in the caption so that it is not necessary to go back to Table 1 to see what it is. It is also necessary to improve the discussion of results, especially in the microstructure part of the gels. With the suggested corrections, the article can be published.

Ans: Thank you very much. All of the treatment captions are described in all Tables and Figures, so it is not essential to return to Table 1 to find out what it is. The black and white figures were already converted to colored figures. The letters on the bars were repositioned on the error bars as needed. The discussion was also improved throughout, particularly in the microstructure of the gel.

Reviewer 3 Report

The research article is highly interesting, but I would not recommend publishing it in its current format due to several areas that require significant improvement. Here are some specific comments:

·         The scientific content of the article is solid; there are no issues in this regard.

·         Many sentences are missing hyphens. It is advisable to add the necessary hyphens.

·         Several sentences contain a series of three or more words, phrases, or clauses. Consider inserting commas to separate these elements.

·         Some sentences include unnecessary instances of "of" and "to."

·         Many sentences lack a comma after the introductory phrase.

·         Numerous sentences are missing proper article usage.

·         Some sentences have unnecessary articles.

·         The authors should be mindful of their use of "was" and "were" in the sentences.

·         The grammar and English throughout the manuscript need comprehensive revision.

The research article is highly interesting, but I would not recommend publishing it in its current format due to several areas that require significant improvement. Here are some specific comments:

·         The scientific content of the article is solid; there are no issues in this regard.

·         Many sentences are missing hyphens. It is advisable to add the necessary hyphens.

·         Several sentences contain a series of three or more words, phrases, or clauses. Consider inserting commas to separate these elements.

·         Some sentences include unnecessary instances of "of" and "to."

·         Many sentences lack a comma after the introductory phrase.

·         Numerous sentences are missing proper article usage.

·         Some sentences have unnecessary articles.

·         The authors should be mindful of their use of "was" and "were" in the sentences.

·         The grammar and English throughout the manuscript need comprehensive revision.

Author Response

Reviewer#3

Comments and Suggestions for Authors

The research article is highly interesting, but I would not recommend publishing it in its current format due to several areas that require significant improvement. Here are some specific comments:

  • The scientific content of the article is solid; there are no issues in this regard.
  • Many sentences are missing hyphens. It is advisable to add the necessary hyphens.
  • Several sentences contain a series of three or more words, phrases, or clauses. Consider inserting commas to separate these elements.
  • Some sentences include unnecessary instances of "of" and "to."
  • Many sentences lack a comma after the introductory phrase.
  • Numerous sentences are missing proper article usage.
  • Some sentences have unnecessary articles.
  • The authors should be mindful of their use of "was" and "were" in the sentences.
  • The grammar and English throughout the manuscript need comprehensive revision.

Ans: All of the English flaws highlighted above were thoroughly double-checked using QuillBot, a paraphrase tool. Prof. Hasene Keskin also proofread the English. Based on the suggestions of all reviewers, a revision was done, and the amended version of the text addressed both academic and English concerns.

Comments on the Quality of English Language

The research article is highly interesting, but I would not recommend publishing it in its current format due to several areas that require significant improvement. Here are some specific comments:

  • The scientific content of the article is solid; there are no issues in this regard.
  • Many sentences are missing hyphens. It is advisable to add the necessary hyphens.
  • Several sentences contain a series of three or more words, phrases, or clauses. Consider inserting commas to separate these elements.
  • Some sentences include unnecessary instances of "of" and "to."
  • Many sentences lack a comma after the introductory phrase.
  • Numerous sentences are missing proper article usage.
  • Some sentences have unnecessary articles.
  • The authors should be mindful of their use of "was" and "were" in the sentences.
  • The grammar and English throughout the manuscript need comprehensive revision.

Ans: All of the English flaws highlighted above were thoroughly double-checked using QuillBot, a paraphrase tool. Prof. Hasene Keskin also proofread the English. Based on the suggestions of all reviewers, a revision was done, and the amended version of the text addressed both academic and English concerns.

Reviewer 4 Report

manuscript is well presented, interesting, more for the industry  and production, than scientific community. I would add "protein content" to the key words.

Author Response

Reviewer#4

manuscript is well presented, interesting, more for the industry  and production, than scientific community. I would add "protein content" to the key words.

Ans: Thank you very much. “protein content” was added in the keywords.

Reviewer 5 Report

I am much honored to have been allowed to review the manuscript. The manuscript entitled "Surimi Production from Tropical Mackerel: A Simple Washing Strategy for Better Utilization of Dark-Fleshed Fish Resource” was reported interesting results. The idea of the article is good. It is almost well designed. However, some points to improve the current format of the article will be mentioned below:

Abstract:

The abstract is insufficient. It is necessary to provide more information. The abstract should be more informative by giving real results rather than elastic sentences. Important and main contents should be given. Support the results with some quantitative data. What is the conclusion?

Introduction:

Line 108-111: The biochemical features, myoglobin and associated species contents, lipid content and oxidative stability, as well as gelling characteristics of the resulting surimi, were extensively evaluated in order to finalize a suitable washing solution for straightforward mackerel surimi production. -> None of these features have been mentioned in materials and methods. Add these features in the article.

Materials and Methods:

Line 166-169: Explain the texture analysis method clearly.

The yield determination method is not mentioned in materials and methods. Please mention it.

Conclusion: what is the future of your findings? Conclusion is not insightful, what are suggestions?

Minor editing of English language required

Author Response

Reviewer#5

Comments and Suggestions for Authors

I am much honored to have been allowed to review the manuscript. The manuscript entitled "Surimi Production from Tropical Mackerel: A Simple Washing Strategy for Better Utilization of Dark-Fleshed Fish Resource” was reported interesting results. The idea of the article is good. It is almost well designed. However, some points to improve the current format of the article will be mentioned below:

Abstract:

The abstract is insufficient. It is necessary to provide more information. The abstract should be more informative by giving real results rather than elastic sentences. Important and main contents should be given. Support the results with some quantitative data. What is the conclusion?

Ans: An abstract was revised accordingly. The quantitative results were included to provide more information and to highlight the important points. The conclusion was also included in the Abstract.

Mackerel (Auxis thazard), a tropical dark-fleshed fish, is currently a viable resource for the manufacture of surimi, but the optimum washing procedure for a more efficient use this particular species is required right away. Washing is the most critical stage in surimi production to ensure optimal gelation with odorless and colorless surimi. The goal of this study was to set a simple washing medium to the test for making mackerel surimi. Washing was done three times with different media. T1 was washed with three cycles of cold carbonated water (CW). T2, T3, and T4 were washed once with cold CW containing 0.3, 0.6, or 0.9% NaCl, followed by two cycles of cold water. T5, T6, and T7 were produced for three cycles with CW containing 0.3, 0.6, or 0.9% NaCl. For comparison, unwashed mince (U) and conventional surimi washed three times in cold tap water (C) were employed. The maximum yield (62.27%) was obtained by washing with T1. When varying quantities of NaCl were mixed into the first washing medium (T2-T4), the yield decreased with increasing NaCl content (27.24-54.77%). When washing with NaCl for three cycles (T5-T7), the yield was much decreased (16.69-35.23%). Conventional surimi washing (C) produced a yield of roughly 40%, which was comparable to T3. Based on the results, treatments that produced lower yields than C were eliminated in order to maximize the use of fish resources and for commercial reasons. The maximum NaCl content in CW can be set at 0.6% only during the 1st washing cycle (T3). Because of the onset of optimal unfolding as reported by specific biochemical characteristics such as Ca2+-ATPase activity (0.2 mmol inorganic phosphate/mg protein/min), reactive sulfhydryl group (3.61 mol/108 g protein), and hydrophobicity (64.02 µg of bromophenol blue bound), T3 washing resulted in surimi with the greatest gel strength (965 g.mm) and water holding capacity (~ 65%), with fine network structure visualized by scanning electron microscope. It also efficiently removed lipid (~80% reduction), myoglobin (~65% reduction), non-heme iron (~94% reduction), and trichloroacetic acid-soluble peptide (~52% reduction) contents, which improves whiteness (~45% improvement), reduces lipid oxidation (TBARS value < 0.5 mg malondialdehyde equivalent/kg), and decreases the intensity of gel's fishy odor (~30% reduction). As a result, washing mackerel surimi (A. thazard) with CW containing 0.6% (w/v) NaCl in the 1st cycle, followed by two cycles of cold water washing (T3), can be a simple method for increasing gel-forming capability and oxidative stability. The mackerel surimi produced using this washing approach has a higher quality than that produced with regular washing. This straightforward method will enable the sustainable use of dark-fleshed fish for the production of surimi.

Introduction:

Line 108-111: The biochemical features, myoglobin and associated species contents, lipid content and oxidative stability, as well as gelling characteristics of the resulting surimi, were extensively evaluated in order to finalize a suitable washing solution for straightforward mackerel surimi production. -> None of these features have been mentioned in materials and methods. Add these features in the article.

Ans: Biochemical features were originally mentioned in Section “3.2. Biochemical Features” with the 2 Subsections “3.2.1. pH” and “3.2.2. Reactive SH, Ca2+-ATPase Activity, and Surface Hydrophobicity”. Also, other parameters had already been included in the Materials and Methods section.

Materials and Methods:

Line 166-169: Explain the texture analysis method clearly.

Ans: The information about the texture analysis was added. “The gel strength (breaking force ´ deformation) of surimi and unwashed mince gels was reported after the texture was evaluated using a TA-XT2 texture analyzer (Stable Micro Systems, Godalming, Surrey, UK) mounted with a spherical plunger (diameter 5 mm; depression speed 60 mm/min/trigger force 0.05 N) [7].

The yield determination method is not mentioned in materials and methods. Please mention it.

 Ans: The yield determination was placed in Section 2.2.1. Measurement of Yield, Moisture, and pH

The yield of surimi from all washing procedures was calculated using the weight of the raw fish material in relation to the weight of minced fish.”

Conclusion: what is the future of your findings? Conclusion is not insightful, what are suggestions?

Ans: At the end of Conclusion, the statement was added. “In the future, upscale production of mackerel surimi using the proposed solution can be done to advance its potential application in industry. It is also possible to investigate the impact of frozen storage on the gel-forming ability and oxidative stability of the produced surimi.

Comments on the Quality of English Language

Minor editing of English language required

Ans: Quillbot, a paraphrasing tool, was used to double-check the English.

Reviewer 6 Report

The abstract is written like experimental recipe. Please improve it

Please modify the introduction. Make it objective oriented and improve the novelty. 

Please give a schematic illustration for the experimental recipe

Which type of water was used for washing , tap water , distilled water or deionized water?

What was the effect of NaCl? What other options as alternate to NaCl are available keeping in mind the cost and efficiency? Please add in discussion

In Figure 3, scale bar is not readable, please improve the quality of figure

Please increase novelty of the work.

Please double check the document for spelling and grammar errors

Most of the references are old. Please replace them with recent work

Please double check the document for spelling and grammar errors

Author Response

Reviewer#6

Comments and Suggestions for Authors

The abstract is written like experimental recipe. Please improve it

Ans: The abstract was revised accordingly. The quantitative results were included to provide more information and to highlight the important points. The conclusion was also included in the Abstract.

Please modify the introduction. Make it objective oriented and improve the novelty. 

Ans: The objective-oriented originality was presented in the final paragraph of the introduction.

Thus, the objective of this study was to increase the capacity for myoglobin removal from mackerel mince and to enhance the quality of the resulting surimi by incorporating NaCl into the CW washing medium. This assumption was made on the premise that increasing the ionic strength of the washing medium by incorporating NaCl will aid in improving the leaching capacity of myoglobin during washing of mackerel mince, as recently reported by Wang et al. [18]. To reduce myofibrillar protein loss and yield reduction, the concentration of NaCl to be included with CW, as well as the cycle of washing with NaCl, should be tuned. The biochemical features, myoglobin and associated species contents, lipid content and oxidative stability, as well as gelling characteristics of the resulting surimi, were extensively evaluated in order to finalize a suitable washing solution for straightforward mackerel surimi production.”

Please give a schematic illustration for the experimental recipe

Ans: Figure 1 depicts a schematic illustration of surimi production and analysis.

Which type of water was used for washing , tap water , distilled water or deionized water?

Ans: It was “cold tap water”. It was added in the text already.

What was the effect of NaCl? What other options as alternate to NaCl are available keeping in mind the cost and efficiency? Please add in discussion

Ans: The effect of NaCl was originally stated in the discussion that “Additional salts, particularly NaCl, are usually employed during the processing to enhance the flavor, color, texture, and shelf-life of muscle-based foods. To solubilize the myofibrillar proteins and attain the appropriate functional characteristics, 2-3% (0.47-0.68 M) NaCl is usually required [35]. Although the salt concentration in the washing media was maintained to no more than 0.9% to prevent myofibrillar protein loss, some myofibrillar proteins were still eliminated during washing, especially with higher salt content and the presence of salt in the 1st and 2nd washing cycles. The DLVO theory, which claims that colloidal stability is determined by a compromise of attractive van der Waals forces and repulsive double-layer electrostatic bonds, can be used to explain the impact of ions on the solubility of protein in colloidal research [36]. Protein can be thought of as a macroion. Under a salt solution environment, it is enclosed by more counterions than coions, which shields the protein surface charge and allows it to solubilize. As the salt content rises, the protective role of protein charge causes the electric double layers to compress and the repulsive term of the DLVO theory to diminish [36]. Protein solubility is projected to reduce at high salt concentrations [37, 38]. Based on the results, treatments that produced lower yields than conventional washed surimi (C) were eliminated in order to maximize the use of fish resources and for commercial reasons. The maximum NaCl content in CW can be set at 0.6% only during the first washing cycle (T3).

However, due of the association among myoglobin and muscle components, particularly myofibrillar proteins, it is difficult to separate all myoglobin from mackerel mince [5]. When washed with the conventional process (C), the myoglobin content decreased by 46.76%. However, CW washing (T1) only removed 3.8% of the myoglobin. This was consistent with a previous study in which CW washing enhanced the gel functionality of mackerel surimi but did not increase heme protein removal [17]. When salt was added to the CW medium, the myoglobin removal efficacy increased to 36.68% for T2 and 64.63% for T3. The presence of salt at the optimal level may have aided in weakening the association between myoglobin and myofibrillar proteins or muscle components, hence facilitating myoglobin leaching efficacy. As previously indicated, raising the ionic strength of the washing medium can improve the solubility of myofibrillar proteins. This can cause myoglobin to dissociate from the myoglobin-myofibrillar complex, allowing myoglobin to be eliminated within the washing water. Unwashed mince had a non-heme iron level of 50.63 mg/g, which significantly decreased following washing with all available methods (Fig. 2a). Non-heme iron was eliminated 93.72% by T3, compared to 79.68, 68.56, and 65.83% by T2, T1, and C, respectively. T3 may remove the non-heme iron through washing media due to the enhanced solubility of myofibrillar proteins at high ionic strength. Overall, the removal of myoglobin and non-heme iron from mackerel by T3 appeared to be the most successful washing procedure, which may have enhanced the gel's functionality and oxidative stability.”

In addition, the new statement was added in the discussion that “The application of NaCl in the washing solution was chosen because NaCl, one of the least expensive substances, is a critical component and a processing aid for a variety of food products [31]. Other NaCl alternatives can be researched in the future to increase the yield and quality of mackerel surimi. However, the proposed compounds' cost and efficiency should be carefully assessed.”

In Figure 3, scale bar is not readable, please improve the quality of figure

Ans: The scale bar is green and was originally from the SEM instrument, which cannot be changed. It can, however, be read by enlarging it with the zoom-in tool. We sincerely sorry.

Please increase novelty of the work.

Ans: The introduction and discussion were rewritten to enhance the novelty of the work as suggested. Thank you very much.

Please double check the document for spelling and grammar errors

Ans: The spelling and grammar errors were double checked using Quillbot.

Most of the references are old. Please replace them with recent work

 Ans: Some old references were replaced with new ones. However, some old references are significant since they contain theoretical data, methods, and supporting materials for the proposed mechanisms that must be maintained.

Comments on the Quality of English Language

Please double check the document for spelling and grammar errors

Ans: Quillbot, a paraphrasing tool, was used to double-check the English.

Round 2

Reviewer 1 Report

The revised manuscript has addressed the issues proposed by the reviewer. Now it can be accepted in this current version.

Reviewer 3 Report

The authors have taken into consideration the most comments/suggestions of the reviewers during the revision of the manuscript, but some minor of grammar and English throughout the manuscript need revision.

The authors have taken into consideration the most comments/suggestions of the reviewers during the revision of the manuscript, but some minor of grammar and English throughout the manuscript need revision.

Reviewer 5 Report

Most of the corrections are done correctly.

Reviewer 6 Report

The authors have addressed the comments well. I have no objection in accepting the article.